# *Aedes albopictus* bionomics data collection by citizen participation on Procida Island, a promising Mediterranean site for the assessment of innovative and community-based integrated pest management methods

Beniamino Caputo[1], Giuliano Langella[2], Valeria Petrella[3], Chiara Virgillito[1,4], Mattia Manica[4,5], Federico Filipponi[1,6], Marianna Varone[3], Pasquale Primo[3], Arianna Puggioli[7], Romeo Bellini[7], Costantino D'Antonio[8], Luca Iesu[3], Liliana Tullo[3], Ciro Rizzo[3], Annalisa Longobardi[3], Germano Sollazzo[3], Maryanna Martina Perrotta[3], Miriana Fabozzi[3], Fabiana Palmieri[3], Giuseppe Saccone[3], Roberto Rosà[4,9], Alessandra della Torre[1], Marco Salvemini[3]*

1 Department of Public Health and Infectious Diseases, University of Rome La Sapienza, Rome, Italy, 2 Department of Agriculture, University of Naples Federico II, Naples, Italy, 3 Department of Biology, University of Naples Federico II, Naples, Italy, 4 Department of Biodiversity and Molecular Ecology, Edmund Mach Foundation, San Michele all'Adige, Italy, 5 Center for Health Emergencies, Bruno Kessler Foundation, Trento, Italy, 6 Istituto Superiore per la Protezione e la Ricerca Ambientale, Rome, Italy, 7 Centro Agricoltura Ambiente "Giorgio Nicoli", Crevalcore, Italy, 8 Ministry of Education, University and Research, Rome, Italy, 9 Centre Agriculture Food Environment, University of Trento, San Michele all'Adige (TN), Italy

* marco.salvemini@unina.it

## Abstract

In the last decades, the colonization of Mediterranean Europe and of other temperate regions by *Aedes albopictus* created an unprecedented nuisance problem in highly infested areas and new public health threats due to the vector competence of the species. The Sterile Insect Technique (SIT) and the Incompatible Insect Technique (IIT) are insecticide-free mosquito-control methods, relying on mass release of irradiated/manipulated males, able to complement existing and only partially effective control tools. The validation of these approaches in the field requires appropriate experimental settings, possibly isolated to avoid mosquito immigration from other infested areas, and preliminary ecological and entomological data. We carried out a 4-year study in the island of Procida (Gulf of Naples, Italy) in strict collaboration with local administrators and citizens to estimate the temporal dynamics, spatial distribution, and population size of *Ae. albopictus* and the dispersal and survival of irradiated males. We applied ovitrap monitoring, geo-spatial analyses, mark-release-recapture technique, and a citizen-science approach. Results allow to predict the seasonal (from April to October, with peaks of 928–9,757 males/ha) and spatial distribution of the species, highlighting the capacity of *Ae. albopictus* population of Procida to colonize and maintain high frequencies in urban as well as in sylvatic inhabited environments. Irradiated males shown limited ability to disperse (mean daily distance travelled <60m) and daily survival estimates ranging between 0.80 and 0.95. Overall, the ecological characteristics of the island, the acquired knowledge on *Ae. albopictus* spatial and temporal distribution, the high human

**Data Availability Statement:** All relevant data are within the manuscript and its Supporting Information files.

**Funding:** This work was supported by two grants "Finanziamento Straordinario del Rettore dell'Università degli Studi di Napoli Federico II" to MS in 2016 and 2017, by a grant from Comitato di Gestione della Riserva Naturale Statale Isola di Vivara to MS in 2018 and by EU funded projects H2020-MSCA-NIGHT-2018 and H2020-MSCA-NIGHT-2020. The funders had no role in study design, data collection and analysis, decision to publish, or preparation of the manuscript.

**Competing interests:** The authors have declared that no competing interests exist.

and *Ae. albopictus* densities and the positive attitude of the resident population in being active parts in innovative mosquito control projects provide the ground for evidence-based planning of the interventions and for the assessment of their effectiveness. In addition, the results highlight the value of creating synergies between research groups, local administrators, and citizens for affordable monitoring (and, in the future, control) of mosquito populations.

## Author summary

Mosquitoes represent a nuisance as well as public health risk due to their ability to transmit pathogens to humans. The Asian tiger mosquito *Aedes albopictus* is an invasive species nowadays established in all Mediterranean countries, reaching highest abundance in Italy. Innovative control approaches have been proposed to complement conventional ones and to increase the success of the fight against this aggressive day-biting species, but still need to be validated in the field. Small islands are ideal places for these validation experiments as they are naturally protected by the spill over of mosquitoes from neighbouring untreated areas. The island of Procida, in the gulf of Naples (Italy), has the right ecological characteristics (e.g., homogeneous landscape and high densities of human and *Ae. albopictus* populations) to represent an ideal experimental site. In collaboration with the Municipality and the residents we obtained relevant data on the mosquito distribution and seasonality on the island and performed preliminary experiments, creating an instrumental baseline information which will facilitate the effective testing of novel control schemes.

## Introduction

The Asian tiger mosquito *Aedes albopictus* (Skuse) (Diptera: Culicidae) is an invasive species that in the last few decades has greatly expanded its range from southeast Asia to all other continents except Antarctica, mostly by the passive transportation of its drought-resistant eggs via used tire trade [1]. Since the first identification in Albania in 1979 [2], this exotic species invaded several European countries thanks to its ability to adapt to seasonal variations and to use man-made water containers for oviposition. Italy is one of the most heavily infested countries in Europe. *Ae. albopictus* was first reported in 1990 in Genova (northwest Liguria region) [3] and quickly spread over the whole territory, in particular in the northeast area and central and southern coastal areas, including major and minor islands [4,5]. *Ae. albopictus* represents a relevant public health risk due to its vector competence for pathogens such as arboviruses and filarial worms [6]. In European Mediterranean regions it has been involved in the last decade in various autochthonous transmission events of chikungunya virus (CHKV) (France [7]), dengue virus (DENV) (Spain [8], Croatia [9] and France [10]) and, more recently, of Zika virus (ZIKV) (France [11]). In Italy in 2007, *Ae. albopictus* was the primary vector during the first European chikungunya outbreak, with >200 human cases in northeast Italy (Emilia Romagna) [12,13]. Ten years later, a second chikungunya outbreak occurred in central and south Italy (Lazio and Calabria), with >500 cases, including cases in the metropolitan city of Rome [14,15] and in a coastal village of south Italy (Calabria) [14].

Mosquito spreading is contrasted with several different methods including microbial larvicides or chemical pesticides. The negative impact of the overuse of chemical compounds on the environment and on human health, the problem of growing insecticide resistance in

mosquito populations including *Ae. albopictus* [16–18], and the absence of available vaccines against most arboviruses, render the development of eco-sustainable alternative mosquito-control methods an urgent need, as recently underlined by WHO [19].

A promising experimental complementary methods for mosquito control are represented by the Sterile Insect Technique (SIT) [20] and the Incompatible Insect Technique (IIT) [21]. SIT relies on mass rearing and mass release of radiation-sterilized males into target areas to suppress local infesting populations. SIT technique has been successfully applied in the frame of area-wide integrated pest management (AW-IPM) programs [22], against several insect species including agricultural pests (*Ceratitis capitata*, *Cydia pomonella*), livestock pests (*Cochliomyia hominivorax*) and vector species such as the tsetse fly *Glossina austeni* [23–26]. SIT potential against mosquitoes has been experimentally demonstrated for the first time in Italy with a three-year long study based on the release of 2 million sterile males of *Ae. albopictus* [27]. By releasing 896–1,590 sterile males/ha/week in small villages it was possible to induce egg sterility in the range 18.72–68.46% causing a significant decline in the egg density [28,29]. From these encouraging pilot studies, the development of experimental SIT tests against mosquitoes has rapidly advanced in recent years thanks to the research and coordinating efforts of the Joint Food and Agriculture Organization of the United Nations (FAO)/International Atomic Energy Agency (IAEA) Insect Pest Control Subprogramme and their collaborators, involved in the development of mass rearing devices, sexing systems and protocols for field evaluation for both *Anopheles* and *Aedes* species [30,31]. IIT relies on mass rearing and mass release of males depleted of their natural *Wolbachia* endosymbiont and harbouring a *Wolbachia* strain different from that present in the wild target mosquito populations. *Wolbachia* is a gram-negative bacterium, a common symbiont of insects, including mosquitoes [32], capable of inducing cytoplasmic incompatibility, i.e., mating between *Wolbachia*-infected males and wild females, without or with a different *Wolbachia* strain, results in embryonic lethality [33]. Pilot studies carried out in Kentucky (USA) and Italy support IIT as a valuable approach to suppress *Ae. albopictus* populations [34,35]. Combining SIT and IIT and releasing millions of factory-reared *Ae. albopictus* adults over a two-year period on two small islands in Chinese city of Guangzhou, Zheng and colleagues have recently demonstrated that the near elimination of field populations of a mosquito specie is achievable [36].

More than thirty SIT or SIT/IIT pilot trials are ongoing worldwide, mainly against *Aedes* mosquitoes [37]. Recently, IAEA and WHO released a guidance framework for the assessment of the feasibility of SIT as a mosquito-control tool against *Aedes*-borne diseases [38] and a phased conditional approach guideline for mosquito management using SIT [37]. According to these guidelines, testing and implementation of Integrated Pest Management (IPM) techniques for mosquito control, including a SIT and/or IIT components, require, as essential premises, i) the selection of a proper and, possibly geographically isolated, site, ii) the baseline data on the bionomics of the local infesting mosquito populations and iii) the local community engagements [37]. In particular, a strong support and commitment of the local community and the early engagement of stakeholders, including residents in study sites and policy makers, are considered essential for the success of field trials [38].

The island of Procida, in the Gulf of Naples (Italy), offers a unique combination of features ideal for the testing of innovative IPM approaches against *Ae. albopictus*. First, the ecological and demographic features: i) a small size (only 4.1 km$^2$), ii) a completely urbanized and accessible territory, iii) a high abundance of small water deposits (e. g., containers in private gardens), iv) a high human population density (27.83 inhabitants/ha—ISTAT 31/12/2020), and v) a history of infestation by *Ae. albopictus*, in the last ten years. Second, the willingness of citizens to participate to the project. Anecdotal evidence collected on the island show that many Procida citizens are familiar with SIT and its effectiveness in insect pest control programs.

This is mainly due to field performance tests of sterilized males of the Mediterranean fruit fly *Ceratitis capitata*, performed in the island during 1970's and 1980's, in a cooperative programme between the Italian National Committee for Research and Development of Nuclear Energy (ENEA) and the IAEA [39,40]. About 20 million sterile Mediterranean fruit fly males were released on the island from April to July 1986 leading to population suppression and protection of citrus fruits and therefore to a very positive perception of SIT by the residents thereafter.

In this paper, we tested a community-based approach to collect baseline data about *Ae. albopictus* bionomics on Procida. Here, we present the results of a four-year project, successfully implemented in strict collaboration with local administrators and citizens. We were able to collect baseline data on the temporal dynamics and the spatial distribution of *Ae. albopictus* on the island; to estimate the *Ae. albopictus* wild population size; and to estimate the dispersal and survival rates of irradiated *Ae. albopictus* males. These data provide insights on the bionomics of the Asian tiger mosquito in southern Europe and draw attention to Procida Island as an ideal site for testing innovative and community-based control programs against *Ae. Albopictus*. The island can be a study model to promote future implementation in other Mediterranean and temperate areas.

## Methods

### Ethics statement

This research was approved by the University of Naples Federico II CSV Ethics Board (Protocol # PG/2020/0090230) and by the Municipality of Procida with municipal resolution n˚52 of 07 July 2016. All procedures performed in studies involving human participants were in accordance with the ethical standards of the institutional and/or national research committee and with the 1964 Helsinki declaration and its later amendments or comparable ethical standards. Informed consent was taken from all the participants involved in the study. Authors of the present manuscript and Procida volunteers gave their written consent to their participation to the study and to be photographed and to have their images published under a creative commons license. All authors declare no competing interests.

### Study site

Procida is a 3.7 km$^2$ flat volcanic island (on average 27 m above sea level) in the gulf of Naples (Italy) with a 16 km-long rough coastline. The Procida territory is uniform in its organization, highly urbanized and completely accessible (S1 Fig). It is organized in hundreds of small residential buildings surrounded by private green areas including ornamental flower and vegetable gardens, orchards with citrus plants and family-type chicken and rabbit shelters. This landscape is associated with an extensive presence of anthropic water sources, representing ideal breeding sites for *Ae. albopictus* larvae. The island has a population density of 27.83 inhabitants/ha (ISTAT 31/12/2020) which approximately doubles during the touristic summer season. A 0.4 km$^2$ satellite island, Vivara, a natural uninhabited reserve, completes the Procida territory. The climate is temperate with an average annual temperature of 16.2˚C and an average annual precipitation of 797 mm (https://en.climate-data.org/) and it is classified as Csa by Köppen and Geiger (Csa = Hot-summer Mediterranean climate) [41].

### Citizen science and community engagement strategy

All experimental field activities presented in this work were performed in strict collaboration with local administrators and citizens of Procida, following the workflow graphically described

in the S2 Fig. The first step was to establish a relationship of mutual trust with the local administration by presenting the *Ae. albopictus* monitoring project, clarifying the logistical support and authorizations needed and drawing a memorandum of understanding between the Municipality and the University of Naples, which was approved with a public deed in July 2016. In parallel to these actions, in September 2015, we assessed the propensity of the island's citizens to volunteer in contributing to our project by administering a questionnaire to about 200 randomly selected Procida residents. The questionnaire included one question to assess the knowledge about the public health relevance of the Asian tiger mosquito, six questions on the preventive measures implemented by people to protect themselves from mosquitoes, and five questions about people possible commitment in control programs against the Asian tiger mosquito. The same questionnaire was administered again in September 2019 to a similar number of randomly selected persons on the island.

Local administrators assisted us in selecting twelve volunteers interested in participating in the *Ae. albopictus* temporal dynamic monitoring effort using ovitraps (S3 Fig). A one week-long workshop, led by expert operators, was organised on site to train the volunteers on how to manage ovitraps and collect germination paper strips to be delivered weekly for egg counting. In April 2016, four teams, composed each of an expert operator, a municipality member and three volunteers, visited the island territory to select twenty-six sites for the ovitrap positioning (S4 Fig). Volunteers managed 13 out of 26 ovitraps form April 2016 to December 2016, while the remaining ovitraps were managed by expert technicians. Then, with a second round of visits to households, we selected 79 families who allowed the access to their private properties to place the ovitraps required for the spatial analysis. 83 households were visited over four days by three teams organized as described previously. Each visit took about 15–20 minute. During the visit, the team members talked with the residents to explain the aim of the study, its relevance for a future control program on the island based on the release of sterile mosquito males and a pamphlet with general information about mosquito biology and control measures was hand-delivered. With a similar approach and supported by a local "facilitator" recommended by Procida administrators, we involved in September 2018 the local community of "La Chiaiolella" area for the mark-release-recapture experiments. Twenty families, two local merchant associations and a parish, which agreed in hosting in their private properties the recapture stations (see below) and two field laboratories for the managing of the sampling instruments and the egg/adult quantification (S3 Fig). The launch of the project in 2016 and the participation of volunteers as citizen scientists during the whole research project period, were covered by various press releases on local and national newspapers, in collaboration with the media office of the Procida municipality (S2 Fig).

During the study period (September 2015 –September 2019), several additional public activities were performed to further promote public awareness and participation of the Procida community to our experiments (S2 Fig). These include: 1) two informative campaigns on the island (October 2017 and September 2018) with hand-distribution of pamphlets about the project progresses and general information about mosquito biology and control measures. 2) one educational activity (May 2019) with fifty students at the secondary school about mosquito biology, monitoring and control. The activity included stereomicroscope observations of life-stages of *Ae. albopictus* in the classroom, in collaboration with science teachers, and homework observations, to be performed under the supervision of the student's parents, of mosquito larvae provided in Falcon tubes to each student. Students were asked to observe and to take photos and drawings of the larvae during its development and to identify the species and the sex of the hatched adults, reporting the results in a brief relation. 3) two public outreach events, held during the European Researchers' Night Week (September 2018 and 2019) as part of the Marie Skłodowska-Curie Actions. Laboratory activities were organized to let citizens to

perform egg counting, sterile males' collections using electric aspirators and observations of mosquitoes marked with fluorescent powder under the stereomicroscope.

## Ovitrap collections for temporal and spatial population dynamics

Cylindrical black plastic jar, 15 cm high, 12 cm in diameter, with an overflow hole at 8 cm from the base, were utilized as ovitraps. Ovitraps were filled with about 600 ml of tap water and walls were lined with heavy-weight seed germination paper strips, 30 x 9 cm in size, (Anchor Paper Co. USA) as oviposition substrate. The strips were lined to the internal wall of the ovitraps to overcome the overflow hole and to prevent possible egg loss due to water overflow. Ovitraps were located at ground level, in shaded areas near the vegetation. For the temporal analysis, 26 ovitraps were distributed all over the island and monitored weekly from 2016-04-14 to 2016-12-31 (S4 Fig and S1 Table). For the spatial analysis on Procida Island, 75 additional ovitraps were deployed across the island and monitored weekly, together with the previous 26, for two weeks in July 2016 (weeks 29–30) and two weeks in September 2016 (weeks 37–38). For the spatial analysis on Vivara Island, 31 ovitraps were located along the two main paths of island and monitored in September 2018 (week 38) and in July 2019 (week 28) (S1 Table and S5 Fig). Collected germination paper strips were brought to the laboratory and eggs counted under a stereomicroscope.

## Production of sterile males

Experimental sterile *Ae. albopictus* male mosquitoes were produced from eggs collected in June 2018 by ovitraps within the study area. Eggs were transported to the laboratory of Sanitary Entomology and Zoology at the Centro Agricoltura Ambiente "Giorgio Nicoli" in Crevalcore (CAA—Bologna, Italy) and utilized to establish a Procida *Ae. albopictus* strain named PRO1. PRO1 strain was reared as previously described [28] in a climate-controlled insectary (28 ± 1˚C, 80 ± 5% RH, 14:10 h L:D photoperiod). CAA is certified with ISO9001, ISO14001 and ISO45001. Larvae obtained after standardized hatching procedures [27] were reared at a fixed larval density (2 larvae/ml) and fed with a standard diet of brewer yeast (IAEA-BY) liquid diet (5.0% w/v) at a mean daily dose of 0.5 mg/larvae [42] for the first four days of development. Pupae were harvested once per cycle at about 24 h from the beginning of pupation and males were separated using a 1,400-micron sieve (Giuliani Tecnologie S.r.l., Via Centallo, 62, 10156 Torino, Italy). With this method, males can be separated out with 99.0% accuracy [43]. However, for this trial particular attention was paid to reduce the female contamination in the released males. Quality control tests carried out on samples of approximately 300 pupae, resulted in 0% of females. Collected male pupae were aged 24 h before being subjected to the irradiation treatments at the Medical Physics Department of St. Anna Hospital (Ferrara, Italy) using a gamma irradiator (IBL 437C, CIS Bio International, Bagnols sur Ceze, France; 65.564 TBq 1772 Ci ± 10% Cs-137 linear source) at a dose of 35 Gy and a dose rate of 2.1 Gy/min (± 3.5%). Following the irradiation procedure, no loss of pupae was observed. Irradiated pupae were then placed in petri dishes (12 cm diameter) filled with water and placed in cardboard boxes (12 x 12 x h 18 cm), supplemented with additional separators to increase the resting areas, closed at the top with mosquito net, for emergence and shipment. Cotton pads soaked with 10% sugar solution were provided and secured at the top of each box. Each box contained 1,500–2,000 adults and provided a vertical resting surface area of 1.3–1.0 cm$^2$/adult. Mosquito boxes were maintained at about 21˚C for two days. After adult emergence the cardboard boxes were transferred inside larger polystyrene container with adequate quantity of gel packs to maintain a constant temperature of 10–15˚C during shipment. About 20,000 adult sterile males were sent by express courier, via ground transportation, in two different expeditions.

## Mark-release-recapture (MRR) experiments

Sterile male releases were performed on 14 (MRR1) and 21 (MRR2) September 2018, at 15:00 PM (40˚45'04.6"N, 14˚00'27.5"E). Temperature and relative humidity at the release sites were 28˚C and 59% and 27˚C and 61% during MRR1 and MRR2, respectively. Immediately before release, males were marked, at room temperature, with a coloured fluorescent powder (PRO-CHIMA s.r.l., Calcinelli di Saltara (PU), Italy, product code PG-661). A fix dose of fluorescent powder (0.3gr/1000 males) was used per each cardboard box and applied, using a manual insufflator (Hygienic vaginal douche, mod. IntimWash, PicSolution, Italy), to disperse the powder uniformly on mosquitoes, following the FAO/IAEA guidelines for Mark-Release-Recapture procedures of *Aedes* mosquitoes [44]. This marking procedure should ensure a successful capture of marked individuals up to 17 days from the release as shown also in Marini et al., 2010 [45]. Fluorescent dust coverage on male body parts was evaluated on samples of about 100 mosquitoes randomly collected from cardboard boxes upon each release. The use of an UV light source was employed to facilitate the identification of dust on the collected male mosquitoes. A purple and green dye was used in MRR1 and MRR2, respectively, to differentiate males of the two releases. Dusted 5 days-old sterile males were released by placing and opening the cardboard boxes in a sunny area without vegetation to favour dispersal. The cages were gently shaken for about 30 min, to induce the males to exit. The males that remained in the cage after 30 min were counted and deducted from the total. Recaptures began approximately 24 hours after each release and performed daily for 13 consecutive days in MRR1 and for 6 consecutive days in MRR2. Recaptures were performed in 39 sampling stations distributed in four concentric annuli (50 meter-distance from each other) around the release point, located in the touristic district of "La Chiaiolella" (~3.1 stations/hectare), in the southern part of the island (S1 Table). At each sampling station, recaptures were performed by BG-Sentinel traps (operating continuously) and by Human Landing Catches (HLC), during the late afternoon peak of *Ae. albopictus* activity (indicatively from 4:30 to 7:30 PM). BG-Sentinel traps baited with BG-Lure were placed at ground level in shaded locations close to domestic areas. HLC has just been utilized for male *Ae. albopictus* captures in previous works, since males are known to be nearby the human host, attracted by the females [46–49]. HLC were carried out by targeting exclusively adult male and female mosquitoes flying around the operator legs and using locally made battery-powered hand-held electric aspirators for 15 minutes at each station. We did not perform any male mosquito collections within the vegetation. Field-collected mosquitoes were identified using ECDC morphological keys [50] and marked males were detected under stereomicroscope and UV lamp.

## Eco-climatic parameters

A spatial dataset to analyse land cover, geomorphological and climatic variables on Procida Island has been generated using open-source Geographic Information System, specifically GRASS GIS [51] for data processing and spatial analysis and Quantum GIS [52] for spatial analysis and layout generation. Land cover variables were retrieved from supervised classification of digital multispectral aerial imagery collected on 16 June 2016 and 07 May 2011 at 0.5 m spatial resolution (Source: Italian National Geoportal, b), using the methodology described in Manica et al. (2016) [53]. Mapped land cover classes were 'trees', 'grasslands', 'roads/concrete', 'buildings', 'bare soil', 'water bodies', 'seawater'. Two main classes are derived from the land cover classified map: artificial surfaces' (including 'roads/concrete' and 'buildings') and 'natural cover' (including 'wood', 'grassland', 'bare soil'). Topography of Procida Island is described in the Digital Terrain Model (DTM) at 2 m spatial resolution, generated from LiDAR acquisitions (Source: Italian National Geoportal, http://www.pcn.minambiente.it/GN/). The

following additional geomorphological descriptors have been computed from DTM data using GDAL library (GDAL/OGR contributors, 2020): slope, aspect, terrain roughness, Topographic Position Index (TPI) and Terrain Ruggedness Index (TRI). In the context of climatic variables, daily spatial maps at 30 m spatial resolution of air temperature climate variable on Procida has been computed combining *in-situ* measured meteorological data and satellite estimated Land Surface Temperature (LST). LST data represent the estimation of skin temperature detected at earth surface by remote sensing sensor. Meteorological data were downloaded from the 'Ciraccio—INAPROCI2' weather station using the Weather Underground database (https://www.wunderground.com/). LST has been estimated from satellite images acquired by OLI and TIRS sensors aboard LANDSAT-8 satellite due to unavailability of LST estimates from MODIS. LST maps at 30 m spatial resolution were computed using the Plank equation, after estimating brightness temperature and emissivity from LANDSAT-8 satellite spectral bands. A total of 16 cloud free satellite images acquired throughout solar year 2016 have been used to estimate LST, using the Land Surface Temperature Estimation QGIS Plugin [54]. A regression analysis has been performed to transform LST estimates, describing the skin temperature of earth surface objects, to the located above air temperature, measured by the weather station at the same satellite acquisition time. The analysis allowed the creation of a regional regression model, that has been trained only accounting for LST estimates in the pixel corresponding to meteorological station location. Finally, the regression model has been used to estimate spatial maps of daily air temperature in Celsius degrees for year 2016 from *in-situ* meteorological measurements, accounting for the spatial variability described in LST maps.

## Statistical analysis

**Temporal analysis.**  A generalized linear additive mixed model (GAMM) was used to assess the relationship between the number of *Ae. albopictus* eggs collected in the 26 ovitraps monitored weekly from April to December 2016 and meteorological variables. GAMM was applied on the series of collected eggs with the following equation:

$$Eggs_{ij} = NB(\mu_{ij}, \theta), \tag{1}$$

$$E\left(Eggs_{ij}\right) = \mu_{ij}; Var\left(N_{ij}^{h}\right) = \mu_{ij} + \frac{\mu_{ij}^{2}}{\theta}, \tag{2}$$

$$\log(\mu_{ij}) = \beta_0 + \beta_1 Rain_{ij} + \beta_1 Wind_{ij} + f(Temperature_{ij}) + \varepsilon_i, \tag{3}$$

$$\varepsilon_i \sim N(0, If_{trap}^2), \tag{4}$$

$Eggs_{ij}$ is the total number of collected eggs at collection week $j$ in ovitrap $i$ and was assumed to follow a Negative Binomial distribution of mean $\mu_{ij}$ and dispersion parameter $\theta$ (Eqs (1) and (2)). A log link function was considered to model $\mu_{ij}$ as a function of independent variables (Eq (3)). *Rain* is the cumulative precipitation during the week of collection, *Wind* is the average wind speed during the week of collection, *f(Temperature)* is the temperature trend modelled by a first order random walk model and *Temperature* is the average temperature during the week of collection. Given that multiple observations were collected from each ovitrap, ovitrap was considered as random effect (ε, Eq (4)). Penalized complexity priors (U = 0.05; α = 0.05) were used for the random walk model of the temperature trend, a log-gamma distribution was used for the priors of the log-transformed precision of the random effect trap, while normal distributions of mean 0 and precision 0.001 were used for the priors of β parameters.

Finally, all quantitative independent variables were standardized (subtracted their mean value and divided by their standard deviation) [55]. GAMM was fitted in a Bayesian framework using INLA [56] and R version 3.5.1 [57]. Assessment of the statistical assumptions in the model was carried out by autocorrelation function, variogram and graphical inspection of the residuals. Model fit was evaluated by computing the Bayesian p-value for each observation and the conditional predictive ordinate (CPO) using 'leave one out' cross validation. 2016 and 2017 meteorological data was used to predict the mean number of eggs in ovitrap for weekly collection and estimate the start and end of the breeding season. The start and end of the season were defined as the week when the cumulative number of eggs collected exceed the 5% and 95% quantile, respectively.

**Spatial analysis.**   The geo-referenced field data from the 101 ovitraps monitored in July and September 2016 were identified on the projection system UTM Zone 33N with datum WGS84 (EPSG code 32633), relating to the Italian cartographic system. The vector and raster maps were prepared and visualized on the Open-Source Quantum GIS version 2.18.2 Las Palmas [52] software and the spatial statistical analysis was undertaken using interpolation by the kriging method on the Open Source R v3.3.2 [57], gstat package [58]. In geo-statistics, the random field (RF) $Z(u)$ is assumed to be intrinsic second order stationary if the first two moments (i.e., the mean or trend component $m$ and the semi-variance $\gamma(h)$) of the two point RF increments exist and are invariant under translation and rotation within a bounded area $D$ [59,60]:

$$m = E\{Z(u)\}, \tag{5}$$

$$\gamma(h) = \frac{1}{2}E\{[Z(u) - Z(u+h)^2]\}, \tag{6}$$

with theoretically infinite points locations $u(x)$ $D$, and random variables (RV) $Z(u)$ and $Z(u+h)$ separated by the distance vector $h(x)$, where $x$ represents the spatial coordinates $(x_1, x_2)$ $\mathfrak{R}^2$ in our study domain. In ordinary kriging the mean is deemed stationary in the local neighbourhood of locations $u$ and unknown, which brings to the following kriging system in matrix notation:

$$Kl(u) = k, \tag{7}$$

Geostatistical interpolation offers the possibility to use the spatial dependency of the variable under investigation to get on the basis of field observations, the values of that target variable at any unsampled or unknown location over the whole study area–in our case in the area of the Procida and Vivara islands. Kriging weights:

$$l(u) = K^{-1}k, \tag{8}$$

are calculated after the fitting of an experimental variogram through an allowed model variogram, which allows in turn to derive the vector of data to unknown covariance:

$$k = \begin{bmatrix} C(u_1 - u) \\ \vdots \\ C(u_n - u) \end{bmatrix}, \tag{9}$$

The vector of weights:

$$l(u) = \begin{bmatrix} l_1(u) \\ \vdots \\ l_n(u) \end{bmatrix},$$

(10)

is calculated by solving the kriging system in equation [8], where:

$$K^{-1} = (K^T K)^{-1} K^T,$$

(11)

The map of interpolation by ordinary kriging (OK) is calculated for each unknown location $u$ of the grid by iterative linear combination of kriging weights with measurements $z(u_\alpha)$ at sampling locations [59]:

$$z(u) = \sum_{a=1}^{n(u)} l_a(u) z(u_a),$$

(12)

**Mean distance travelled, population survival and size.** Mean distance travelled (MDT) was computed to estimate the dispersal of marked *Ae. albopictus* males from MRR experiments, taking into account the unequal trap densities within each annulus [61,62]. MDT is independent of the position of the traps or size of study area [63–65], and is defined by the following equation:

$$MDT = \frac{\sum_{i=1}^{a} ER_i \gamma_i}{\sum_{i=1}^{a} ER_i},$$

(13)

where, $a$ is the number of annuli ($a = 4$), is the median distance of each annulus, ER is the number of recaptures that would be expected if trap density were constant within annulus. See S1 Text for further details.

Based on MDT results, which provide an estimate of the area to consider in the estimation of the population size, a Generalized Linear Model (GLM) with Binomial distribution was used to estimate the population survival rate of marked *Ae. albopictus* males and the population size of the wild population [66]. GLM was applied on the series of collected marked males out of the total number of collected male mosquitoes with the following equation:

$$log\left(\frac{\pi}{1 - \pi}\right) = -log(N) - \lambda t + log(M),$$

(14)

Where N is the population size, M is the number of mosquitoes released, $\lambda$ is the rate of survival function and t is the time (days) between release and recapture of mark mosquitoes as in Cianci et al. (2013). See S2 Text for further details.

Moreover, the Fisher-Ford's method modified for low recapture rate [66] was also applied to provide a second estimate of population size. The Fisher-Ford's equation is the following:

$$N = \frac{\varphi^t (n+1)(M+1)}{m+1},$$

(15)

where N is the population size, $\varphi$ is the marked male survival function, t is the time between release and capture, n is the number of both marked and unmarked mosquitoes captured, m is the number of marked mosquitoes recaptured and M is the number of marked mosquitoes released. The confidence intervals related to the estimates are calculated with the method of percentile bootstrap [67] based on 1000 bootstrap replicates at 95% level.

## Results

### Community-engagement assessment

Over four years of activities on Procida Island, ~300 residents were involved as citizen scientists or collaborators. Twelve persons, including the Procida mayor himself and two municipal counsellors, contributed as volunteers to the ovitrap monitoring. Ten out of twelve volunteers participated in the ovitrap monitoring for a complete *Ae. albopictus* season, from April 2016 to December 2016. The two remaining volunteers dropped out after three months of activity, because of holidays abroad or loss of interest in the project. About 300 persons, belonging to 99 household families, contributed to the spatial analysis of *Ae. albopictus* distribution on the island and to the mark-release-recapture experiments. In addition, we estimated that ~2,000 people were directly or indirectly contacted during the four years of activity on the island (direct participation as volunteers, household families, students and their families, people receiving informative pamphlets, people participating in surveys and to the public outreach activities). In September 2015 we evaluated, using a questionnaire, the knowledge of Procida citizens about *Ae. albopictus*-associated public health risks, the kind of practices they utilized to protect themselves by mosquitoes on the island and their attitude in supporting and participating in mosquito research and control campaigns (Table 1).

Among the 200 randomly selected people interviewed (~2% of the resident population), 77% resulted aware of the capacity of mosquitoes to transmit diseases, 85% declared to use electric diffusers, mosquito nets or chemical repellents to protect themselves from bites, only 11% removed water containers from their properties and only 3% used larvicide products in standing water. The large majority of interviewed people was in favour of a mosquito control programme on Procida, but a minority of them agreed to contribute economically (33%) or to participate as volunteers in the control programme (25%) [68]. In September 2019, we administered again the same questionnaire to 191 randomly selected residents to record possible changes in feedback after our research activities on the island. While no significant changes were observed in the knowledge about the Asian tiger mosquito as vector and about preventive/protective measures taken (with the only exception of an apparent increase in the use of mosquito nets), we observed an increased interest in mosquito control program and an increased availability in supporting or participating to monitoring and control actions on the island.

**Table 1. Results of public surveys carried out among Procida Island residents in 2015 (N = 200) and 2019 (N = 191) using a questionnaire.**

| Questionnaire question | Responses 2015 (%) | | Responses 2019 (%) | |
|---|---|---|---|---|
| | YES | NO | YES | NO |
| Do you know that the Asian tiger mosquito can transmit viral diseases to humans? | 77 | 23 | 73 | 27 |
| Do you use protective measures against mosquitoes? | 85 | 15 | 91 | 9 |
| Do you use electric diffusers? | 58 | 42 | 37 | 63 |
| Do you use mosquito nets? | 54 | 46 | 71 | 29 |
| Do you use insect repellents? | 45 | 55 | 47 | 53 |
| Do you use larvicides? | 3 | 97 | 2 | 98 |
| Do you remove standing water? | 11 | 89 | 11 | 89 |
| Would you welcome a regional/municipal mosquito control programme? | 88 | 12 | 96 | 4 |
| Would you agree to the installation in your property of traps for the capture and monitoring of mosquitoes? | 44 | 56 | 78 | 22 |
| Would you agree to contribute personally to the financing of a mosquito control project? | 33 | 67 | 54 | 46 |
| Are you interested in participating, as a volunteer, to a mosquito monitoring and control programme in Procida? | 25 | 75 | 33 | 67 |

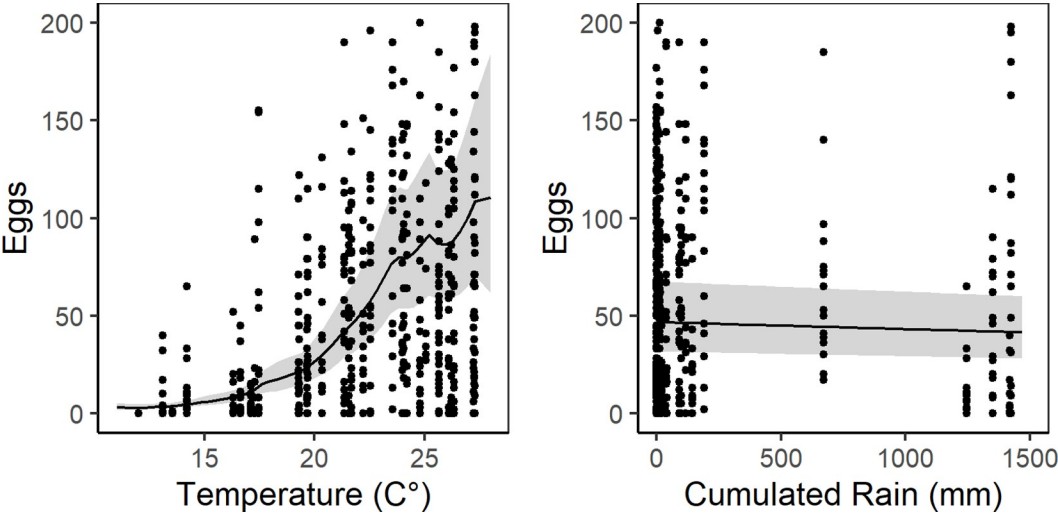

**Fig 1. GAMM posterior predictive values of *Ae. albopictus* eggs/ovitrap/week on Procida Island.** Left panel = temperature dependent mean value of eggs/ovitrap/week. Right panel = rain dependent mean value of eggs/ovitrap/week. Solid lines = GAMM posterior mean value of eggs/ovitrap/week. Grey areas = 95% credible interval. Black dots = observed values of eggs/ovitrap/week. y-axis = number of eggs/ovitrap/week. x-axis (left panel) = weekly averaged temperature. x-axis (right panel) = weekly accumulated precipitation.

## Temporal analysis

A total of 44,244 *Ae. albopictus* eggs were collected by 26 ovitraps located across Procida and Vivara Islands in 2016 and two peaks of egg abundance were registered, one in July and one in September, in most ovitraps (see Methods, S4 Fig and S1 Table). Results of a generalized additive mixed model (GAMM) carried out to model the relationship between meteorological variables and *Ae. albopictus* egg abundance, show evidence of a strong effect of temperature on the number of weekly collected eggs and low or not significant effects of cumulated rain and average wind speed, respectively (Fig 1 and Table 2).

Model assessment shows no-specific violation of statistical assumptions (homogeneity, independence, autocorrelation, spatial correlation) (S6 Fig). However, GAMM shows some under-dispersion (dispersion statistic = 0.71) under the assumption of a Negative Binomial distribution and a strong over-dispersion under a Poisson distribution (dispersion statistic = 39.3). Moreover, the model predicts a lower number of zeros compared to the observed ones (S7 Fig). Model fit estimated by Conditional Predictive Ordinate (hereafter CPO, see

**Table 2. Results for the GAMM.** Dependent variable is the count of eggs in ovitrap, independent variables are wind, rain considered as fixed effect and temperature as spline function, while the position of ovitraps as randomised effect. The posterior mean values and 95% credible intervals for both parameters and hyperparameters are provided. When the 95% credible interval includes zero there is no statistical support of a correlation between the independent and the dependent variable.

| Parameters | Mean (95% credible interval) |
|---|---|
| Intercept ($\beta_0$) | 2.942 (2.633; 3.248) |
| Rain ($\beta_1$) | -0.310 (-0.450; -0.162) |
| Wind ($\beta_2$) | -0.101 (-0.218; 0.019) |
| **Hyperparameters** | |
| Negative binomial size parameter ($1/\vartheta$) | 0.536 (0.474; 0.605) |
| Precision for Temperature random walk model | 6.533 (3.615; 10.831) |
| Precision for ovitrap random effect | 2.447 (1.199; 4.372) |

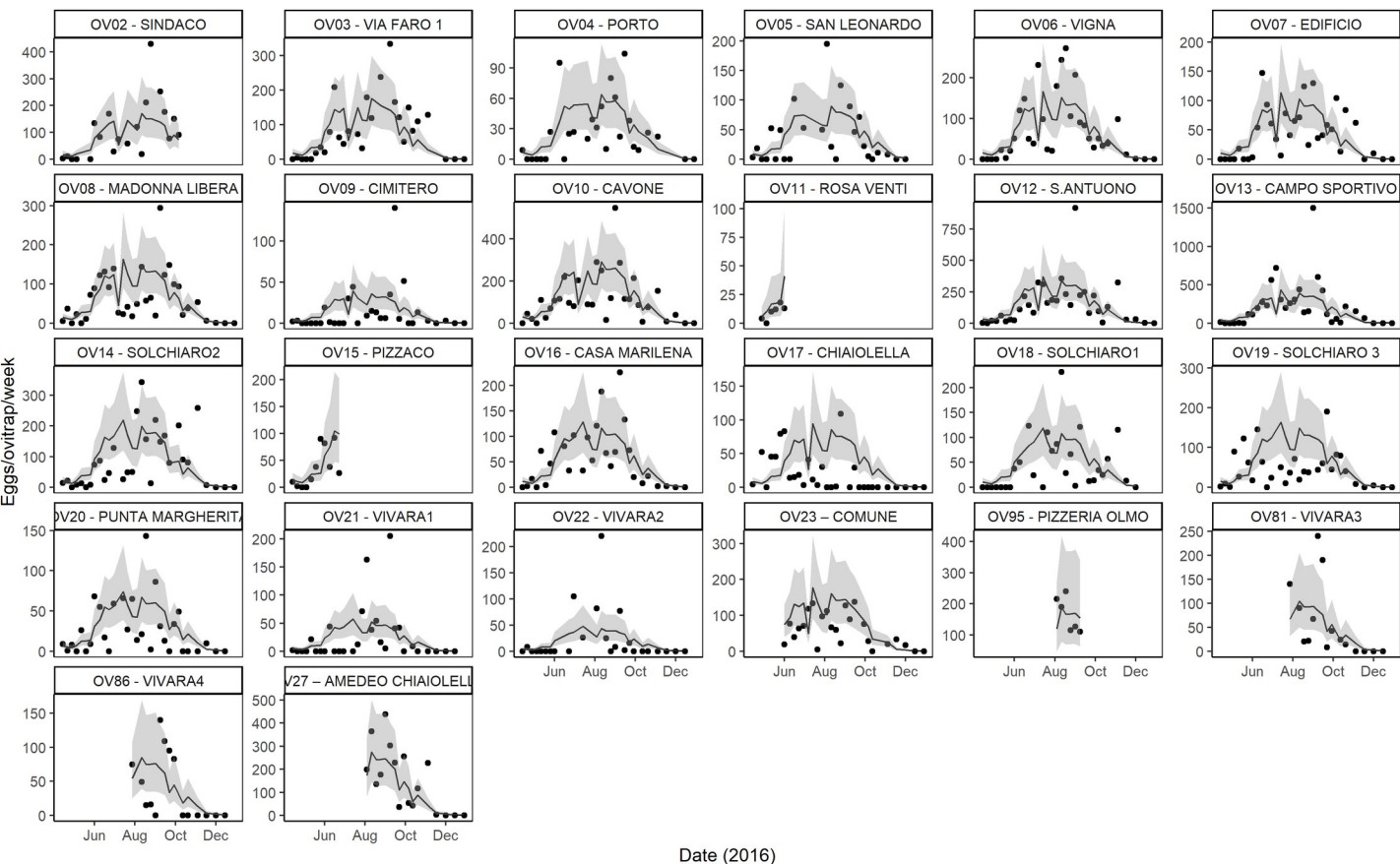

**Fig 2. Observed and expected values *Ae. albopictus* eggs/ovitrap/week on Procida Island in 2016 estimated by GAMM model.** Each panel represents a single ovitrap. Black dots = observed values of eggs/ovitrap/week; solid line = GAMM posterior mean value of eggs/ovitrap/week; grey area = 95% credible interval; x-axis = months of collections in 2016; y-axis = eggs/ovitrap/week; the scale differs per panel to help visualization.

Methods, section Temporal analysis) and Bayesian p-value is poor (S7 Fig), probably due to the large variation detected in ovitrap capture (range: 0–1500) coupled with a considerable proportion of zero captures (~25.7%) (Fig 2). However, fitted values positively correlate with observed values (Pearson's correlation: 0.697, df = 660, p-value <0.001) and the predicted oviposition temporal pattern is overall consistent with the observed one (Fig 2). In particular, the model well-captures the decreasing oviposition activity at the end of the season, but it is challenged by the high variability observed at trap level and during the season (see as an example OV09 or OV12).

## Spatial analysis

A total of 40,811 *Ae. albopictus* eggs were collected by weekly monitoring of 101 ovitraps distributed across Procida in July (weeks 29–30) and September (weels 37–38) 2016. By means of the experimental variogram, we detected the presence of a spatial structure in the data sampled (eggs/ovitraps/week) (Fig 3). A preliminary analysis of the cloud variograms (S8 Fig), showed a good spatial structure in the total number of eggs/ovitraps/week with a relatively high sill-nugget ratio (S9 Fig). This suggests that an important part of the sample variance can be explained by the spatial covariance and can be used, by means of kriging weights, to make spatial interpolation over a regular grid to get a continuous map of eggs deposited in the ovitraps.

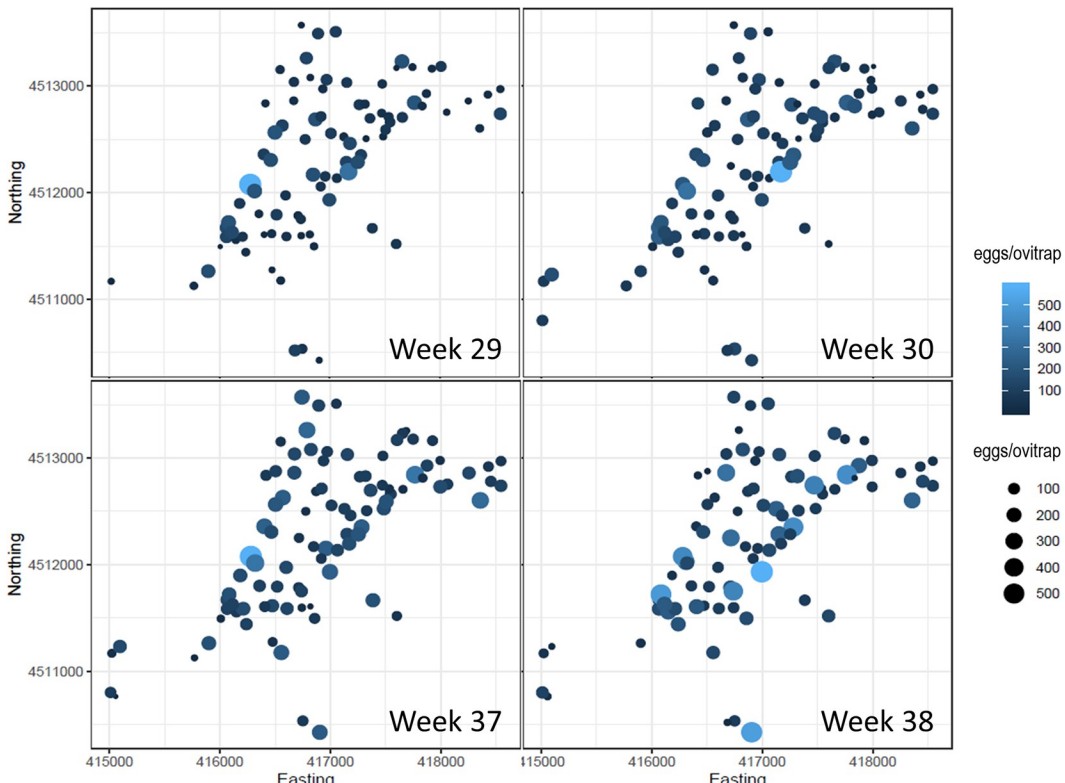

**Fig 3. Number of *Ae. albopictus* eggs/ovitrap/week on Procida Island in July and September 2016.** Spatial coordinates Northing and Easting are expressed in meters using the coordinate reference system with EPSG code 32633 (https://epsg.io/ 32633). Circles represent the positions of the 101 ovitraps utilized for the spatial analysis. Circle size and colour scales are proportional to the magnitude of collected eggs/ovitrap. The double scale for the same parameter enables the resolution of the overlapping circles in the plot.

The ordinary kriging maps highlights a uniform presence of *Ae. albopictus* on the island, with some "hot spots" of oviposition activity (Fig 4). We calculated a set of geospatial covariates that could aid the spatial analysis of target measurements (ovipositions) in a multivariate context. Different data sources were used, such as a digital elevation model, a map of land use and land cover and a map of distance from the sea. Four rasters of growing degree days were calculated, one for each week, as a proxy of the spatial distribution of *Ae. albopictus* ovipositing females. All the tested covariates poorly contribute to the analysis of spatial variance of oviposition, as highlighted by very low Pearson correlation values, ranging from -0.19 to 0.28, and no significant correlations were observed (S2 Table).

## Ovitrap monitoring of *Ae. albopictus* in Vivara reserve

Temporal analysis in 2016 found *Ae. albopictus* also in Vivara, an inhabited natural reserve, where potential hosts are represented by rodents, birds and reptiles, only. To analyse the mosquito spatial distribution in the reserve area, we monitored the oviposition rate by 31 ovitraps in 2018 and 2019 (see Methods, S1 Table and S5 Fig). All ovitraps were found positive for *Ae. albopictus* eggs: a total of 18,891 and 6,608 eggs were captured in September 2018 (611 eggs/ trap/week) and in July 2019 (213 eggs/trap/week), respectively. In addition, we performed adult collections by four BG-sentinel traps baited with BG-lure (S5 Fig). We collected 7 males and 31 females by BG-trap n°2 in September 2018 and 19 males and 27 females by BG-traps n°

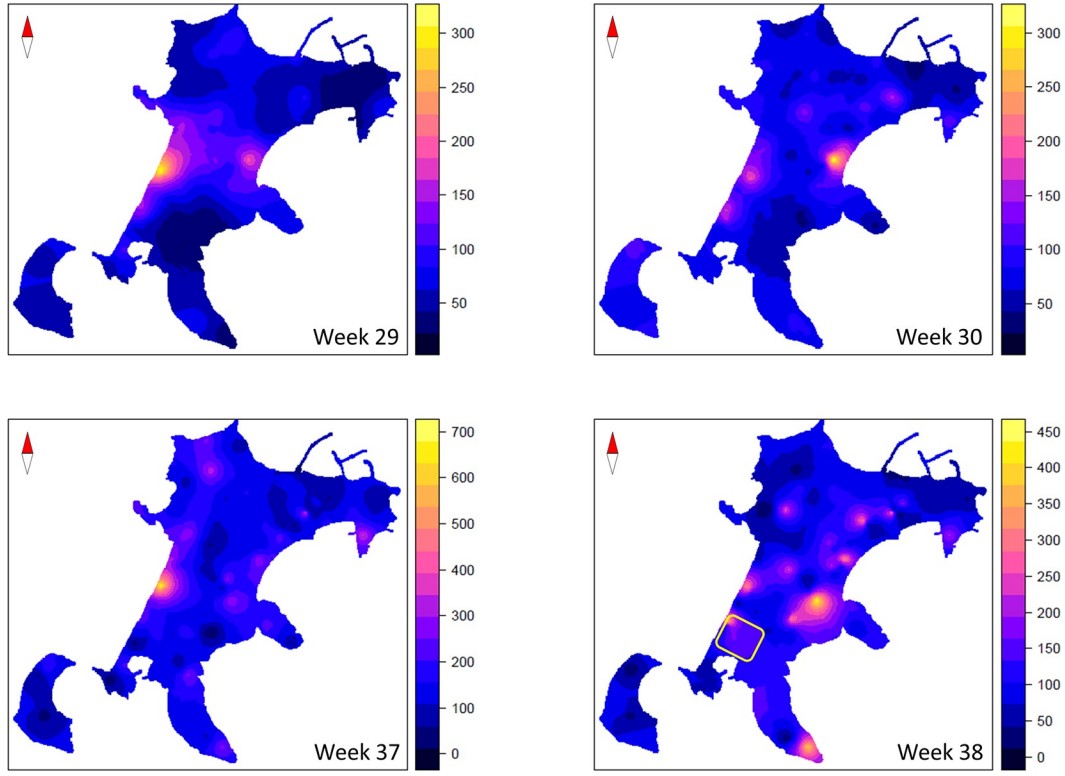

**Fig 4. Ordinary Kriging maps of *A. albopictus* oviposition on Procida and Vivara islands.** The figure shows the estimated ordinary kriging of *Ae. albopictus* oviposition on Procida Island in 4 weeks from July to September 2016. The colour gradient corresponds to the variation range of the estimated egg numbers. The yellow rectangle represents the area selected for the MRR experiments. The base layer of the geographic background map has been sourced from an open maps access (https://glovis.usgs.gov/app).

2 and n°3 in July 2019 (S1 Table). Both positive BG-traps were in proximity of two breeding sites present on the Vivara reserve (S5 Fig).

## Mark-release-recapture experiments

We performed two mark-release-recapture (MRR) experiments in September 2018 (i.e., in the seasonal high-density peak according to results of 2016 spatial analysis) to estimate *Ae. albopictus* male density per hectare and to evaluate the dispersal and survival capacity of males sterilized by radiation (Fig 5).

In total, 7,836 (MRR1) and 9,680 (MRR2) marked sterile *Ae. albopictus* males (PRO1 strain) were released in September 2018. In MRR1, 2,009 wild males and 169 marked males were recaptured by HLC (N = 79) and by BG (N = 90) during 14 days of collection, with an overall recapture rate of 1.8% (N = 143) in the first 6 days. During this time interval, 97.8% of the males were recaptured within 50 m from the release site and none at ≥150 m from it (Table 2). In MRR2, 2,319 wild males and 165 marked males were recaptured by HLC (N = 111) and BG (N = 54) during 6 days of collection, with a total recapture rate of 1.7%. Among marked males, 87.9% and 6.7% were recaptured within 50 m and at ≥150 m distance from the release site, respectively (Table 3). Overall, only 4% and 9% of recaptured males were collected by BG sentinel and HLC beyond 50 m from the release point, respectively (Table 3).

Estimates of cumulative MDTs (1–6 days) was 51 m in MRR1 and 61 m in MRR2 based on BG-trap collections, and 52 m in MRR1 and 57 m in MRR2 based on HLCs (Table 4).

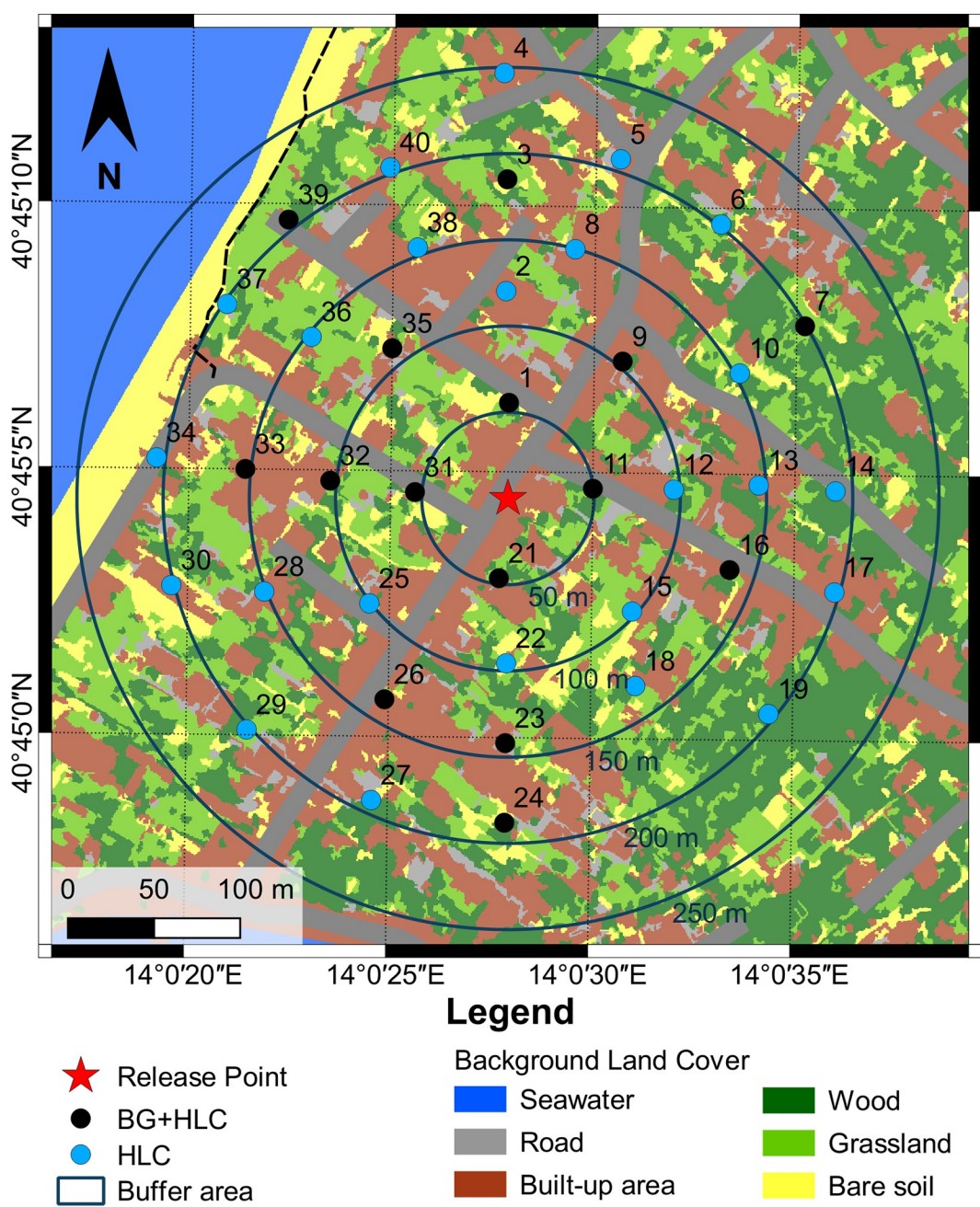

**Fig 5. Distribution of *Aedes albopictus* sampling stations in the mark-release-recapture study site in "La Chiaiolella" (Procida).** Red star = release site. Black circles = HLC + BG-traps recapture stations. Light blue circles = HLC recapture stations. Circles = 50 m-annuli around release site. The base layer of the geographic background map has been sourced from an open maps access (https://glovis.usgs.gov/app).

Population size of wild *Ae. albopictus* males was estimated over areas of either 50 m (estimated flight range) or 200 m (whole sampling area) radius (Table 4 and Fig 6): results predict that the number of males/ha is higher within 50 m-radius area than in the 200 m-radius area. This effect is probably due to heterogeneities in mosquito abundance (as estimated by field collection), which increases at a lower rate than the size of the area (which increases

**Table 3. Number of marked (and unmarked) *Ae. albopictus* males recaptured in two Mark-Release-Recapture (MRR) experiments on Procida Island.** BG = BG-sentinel traps (number of BG within each annulus); HLC = Human Landing Catches (number of HLC sites within each annulus).

| | Annuli (mt) | Trap/ sampling (number) | Days after release | | | | | | | |
| | | | 1d | 2d | 3d | 4d | 5d | 6d | 7d | 8-14d |
|---|---|---|---|---|---|---|---|---|---|---|
| **MRR1** | 0–50 | BG (4) | 6 (35) | 31 (88) | 18 (72) | 16 (51) | 6 (19) | 4 (36) | 3 (32) | 5 (377) |
| | | HLC (4) | 5 (24) | 3 (21) | 17 (68) | 20 (75) | 11 (45) | 6 (44) | // | 14(426) |
| | 50–100 | BG (3) | 0 (28) | 0 (13) | 0 (28) | 0 (40) | 0 (23) | 0 (86) | 0 (36) | 1 (345) |
| | | HLC (8) | 0 (13) | 1 (44) | 0 (51) | 0 (32) | 0 (43) | 1 (38) | // | 0 (225) |
| | 100–150 | BG (5) | 0 (16) | 0 (14) | 0 (20) | 0 (22) | 0 (17) | 0 (10) | 0 (17) | 0 (124) |
| | | HLC (11) | 0 (37) | 0 (48) | 0 (39) | 1 (68) | 0 (74) | 0 (85) | // | 0 (272) |
| | 150–200 | BG (3) | 0 (12) | 0 (9) | 0 (2) | 0 (9) | 0 (5) | 0 (4) | 0 (2) | 0 (46) |
| | | HLC (16) | 0 (58) | 0 (61) | 0 (78) | 0 (119) | 0 (121) | 0 (76) | // | 0 (369) |
| **MRR2** | 0–50 | BG (4) | 2 (63) | 27 (90) | 10 (81) | 5 (70) | 2 (18) | 3 (73) | // | // |
| | | HLC (4) | 11 (104) | 33 (114) | 38 (91) | 10 (56) | 3 (24) | 1 (37) | // | // |
| | 50–100 | BG (3) | 2 (43) | 0 (46) | 1 (57) | 0 (118) | 1 (62) | 0 (19) | // | // |
| | | HLC (8) | 1 (43) | 2 (31) | 0 (52) | 0 (44) | 0 (25) | 0 (60) | // | // |
| | 100–150 | BG (5) | 0 (26) | 0 (30) | 0 (34) | 0 (8) | 1 (10) | 0 (16) | // | // |
| | | HLC (11) | 0 (64) | 0 (57) | 1 (42) | 0 (60) | 0 (24) | 0 (25) | // | // |
| | 150–200 | BG (3) | 0 (12) | 0 (13) | 0 (6) | 0 (8) | 0 (5) | 0 (2) | // | // |
| | | HLC (16) | 0 (93) | 0 (104) | 10 (68) | 1 (40) | 0 (16) | 0 (48) | // | // |

quadratically). Estimates of mosquitoes/ha based on recaptures in the 50 m-radius area largely differ depending between GLM and Fisher Ford methods (Table 5 and Fig 6). The estimated survival parameter of marked mosquitoes ranges from 0.8 to 0.95, according to the sampling area and the trap type considered in the estimation.

## Discussion

The application of sustainable innovations in mosquito control programmes requires strong partnerships with local communities [69,70]. The achievement level of such sustainable control programmes greatly depends on the degree to which citizens are effectively engaged into the work of entomological research and mosquito control, as reminded by WHO's 2017–2030 Global Vector Control Response [71]. At the same time, bionomics data collection for the vector species of interest and the selection of pilot field test sites are considered essential premises for the successful implementation of sustainable vector control programmes, in particular for *Aedes* species [72]. Here, we present bionomics data of *Ae. albopictus* population on the island of Procida, an interesting site for field testing of innovative control methods. The data were

**Table 4. Mean distance travelled (in meters) of recaptured *Aedes albopictus* sterile males in two mark-release-recapture experiments on Procida Island.**

| | MRR1 (BG) | MRR2 (BG) | MRR1 (HLC) | MRR2 (HLC) |
|---|---|---|---|---|
| day after release | | | | |
| 1 | 51 | 94 | 51 | 55 |
| 2 | 51 | 51 | 63 | 57 |
| 3 | 51 | 61 | 51 | 61 |
| 4 | 51 | 51 | 53 | 55 |
| 5 | 51 | 89 | 51 | 51 |
| 6 | 51 | 51 | 58 | 51 |
| 1/6 days | 51 | 61 | 52 | 57 |

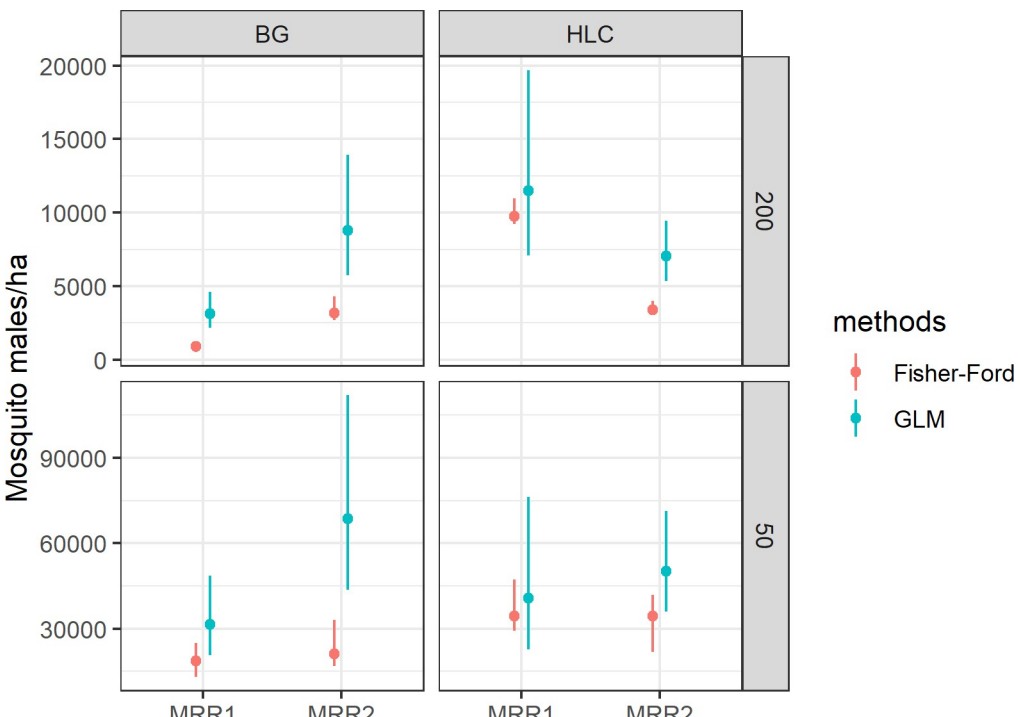

**Fig 6. Wild *Aedes albopictus* population size on Procida Island estimated based on mark-release-recapture (MRR) experiments within either a 200m- and a 50m-areas around release site.** Blue lines = GLM-based estimates; red lines = Fisher-Ford based estimates; dots = mean value of mosquito males/hectare; vertical segments = 95% confidence intervals using percentile bootstrap (red), using equation [5, S1 Text] (blue).

**Table 5. Wild *Aedes albopictus* population size on Procida Island and survival of sterile males estimated based on mark-release-recapture (MRR) experiments within either a 50m- and a 200m-areas around release site.** Density/ha = number of males/ha; N = estimate of population size/day; λ = survival rate estimated by logistic regression; confidence intervals based on 1000 bootstrap replicates with the method of percentile bootstrap at 95% level; m = number of marked mosquitoes recaptured; n = total number of mosquitoes captured (marked+wild).

| MRR | Methods | Type of traps | Annuli (m) | N | Density per ha | n | m | λ | 95% CI per density per ha |
|---|---|---|---|---|---|---|---|---|---|
|  | Regression Model | BG | 50 | 24692 | 31437 | 301 | 81 | 0.92 | 20753–48490 |
| 1 |  |  | 200 | 39361 | 3132 | 659 | 81 | 0.8 | 2173–4618 |
|  |  | HLC | 50 | 32013 | 40761 | 277 | 62 | 0.87 | 22786–76221 |
|  |  |  | 200 | 144447 | 11494 | 1350 | 65 | 0.95 | 7075–19673 |
|  |  | BG | 50 | 14727 | 18750 | 301 | 81 | \\ | 13048–24835 |
|  | Fisher-Ford(a) |  | 200 | 11659 | 928 | 659 | 81 | \\ | 485–963 |
|  |  | HLC | 50 | 27058 | 34450 | 277 | 62 | \\ | 29223–47132 |
|  |  |  | 200 | 122611 | 9757 | 1350 | 65 | \\ | 9301–10989 |
|  |  | BG | 50 | 53928 | 68663 | 395 | 48 | 0.81 | 43566–112028 |
| 2 | Regression Model |  | 200 | 110256 | 8774 | 910 | 53 | 0.8 | 5753–13914 |
|  |  | HLC | 50 | 39481 | 50267 | 426 | 96 | 0.94 | 36021–71275 |
|  |  |  | 200 | 88389 | 7034 | 1322 | 114 | 0.86 | 5333–9425 |
|  |  | BG | 50 | 16618 | 21158 | 395 | 48 | \\ | 16810–31257 |
|  | Fisher-Ford(a) |  | 200 | 339776 | 3165 | 910 | 53 | \\ | 2700–4276 |
|  |  | HLC | 50 | 27024 | 34408 | 426 | 96 | \\ | 21814–41779 |
|  |  |  | 200 | 42421 | 3375 | 1322 | 114 | \\ | 2526–3367 |

collected using a community-based approach, which includes a citizen science component and several community-engagement activities, designed to involve the Procida community from the early stages of our research activities on the island.

## Community-based approach to mosquito monitoring on Procida Island

Citizens have recently been included in citizen science projects for mosquito surveillance and control in several countries all around the globe, such as Germany [73], Spain [74] and USA [75]. In these studies citizens were asked to contribute by reporting the mosquito fauna by submitting mosquitoes collected in their private surroundings [73], by reporting tiger mosquito presence through a mobile app [74], or by reporting about potential mosquito habitat such as larval breeding sites [75]. Recently, Schoener and colleagues reported the involvement of citizens in Austria to monitor *Aedes* species by using ovitraps during summer months, from July to October 2017 [76].

We actively involved Procida citizens in monitoring ovipositing mosquitoes on the island over an entire season, from April to December 2016. To the best of our knowledge, this is the first citizen science project successfully involving volunteers in ovitrap managment for such a long period of time. We observed a drop out of only two volunteers out of twelve during the nine months of ovitrap monitoring. The active participation of the mayor and of the two municipal counsellors as volunteers in the project was an example for the other volunteers and certainly contributed to motivate citizens to participate in the monitoring until the project conclusion. This also represents a novelty in the community-engagement strategy for mosquito control programs that could be further explored in the future.

We also obtained a high success rate in the family involvement in the analysis of spatial distribution of *Ae. albopictus* on the island: 99 out of 106 (93.4%) contacted families gave access to their private properties to place the ovitraps or to set the re-capture stations for the MRR experiments. This provided us with capillary access to the whole island territory, enabling an easy and appropriate spatial distribution of the sampling stations.

Our results clearly show that the effort of the research groups in working in synergy with public administration and citizens of Procida was instrumental to implement high quality monitoring and experimental procedures despite the limited budget available. According to the surveys carried out in 2015 and 2019, Procida citizen involvement in mosquito monitoring and control can be further increased. The limited commitment of the residents in eliminating standing water and in larvicide usage highlights the need to increase knowledge on best practices to be implemented at the individual level to reduce mosquito reproduction (e.g., by specific educational projects in schools and/or door-to-door activities on mosquito biology and control) as a relevant step to lower mosquito densities and creating a condition more suitable for the success of pilot SIT/IIT studies and public mosquito control activities.

## Spatial-temporal population dynamics on Procida Island

Results from ovitrap monitoring allowed to characterize the temporal dynamics and the spatial distribution of the *Ae. albopictus* on the island of Procida and to develop predictive models useful for optimization of conventional mosquito control interventions, as well as for the assessment of the effectiveness of control tools, such as SIT and/or IIT which still need to be validated in the field.

From the seasonal perspective, modelling of 2016 ovitrap and meteorological data allowed to predict the start of *Ae. albopictus* seasonal activity on Procida in April/early May and its end in October/early November (S10 Fig). Egg abundance dynamics is shown to be dependent on temperatures (Table 1 and Fig 1) and is characterized by two peaks in July and September. These

predictions are consistent with a >6 month active season in central Italy [53,77], while the species seasonality is known to be shorter in northern regions [78]. It is important to note that the results obtained by the temporal analysis had a limitation due to the monitoring of the population dynamic for just one year and to the use of the ovitraps, known to have large variation in collections. However, our model results are useful for evidence-based identification of the most appropriate period to start implementing mosquito control interventions on Procida (i.e., not later than April), as well as for assessing the effectiveness control approaches by comparing the predicted natural seasonal dynamics of the species with that observed after interventions. Further field data collection and analyses will allow the validation of the GAMM model, which could represent a trustable predictive model exploiting locally collected meteorological data to define more precisely the starting period of control interventions on Procida Island and in other Mediterranean sites.

To analyse the spatial distribution of the *Ae. albopictus* on Procida Island we applied a geostatistical approach. Geostatistical approaches, with particular emphasis on ordinary kriging, have been previously applied to study spatial distribution of insect vectors, e.g., to study either the association between vector distribution and pathogen transmission [79] and the critical low temperature for the survival of *Aedes aegypti* in Taiwan [80], or to relate entomological indicators to the incidence of dengue [81] and to analyse seasonal and spatial distribution of *Ae. aegypti* and *Ae. albopictus* in San Paulo municipality (Brazil) [82]. Our results highlight an overall uniform and massive presence of the *Ae. albopictus* mosquito consistent with the nature of the island territory, completely urbanized and fragmented in hundreds of private properties with residential buildings surrounded by gardens. Indeed, it was very easy to identify several anthropic water sources ideal for the species larval breeding in all sites selected for ovitrap positioning. In the hypothesis of the release of males within SIT and/or IIT plans, this finding points to the need of an island-wide capillary release of males (e.g., by drones [83] and/ or by a large network of volunteers involved in ground releases) in order to achieve an effective and long lasting suppression trial. Furthermore, the uniformity of the Procida territory and the observed uniform and massive presence of the *Ae. albopictus* mosquito make easy the identification of island's areas to be utilized as test and reference field sites for a robust testing of SIT and/or IIT, as recently suggested by Oliva and colleagues [72].

As for the main island, a high *Ae. albopictus* densities have been also observed in the natural reserve of Vivara. This finding confirms that also in the Mediterranean area, as shown by several recent studies in invaded countries of the New-World [84–86], the species maintains the capacity to colonize sylvatic environments. In addition, this observation highlights the importance of also considering these kinds of habitats for the proper planning of future mosquito control interventions on Procida.

## Procida *Ae. albopictus* population size estimates

We estimated *Ae. albopictus* population size at the end of the reproductive season (September) by MRR experiments. Estimates based on 200m-dispersal (928–9,757 males/ha with Fisher-Ford and 3,132–11,494 males/ha with regression) are comparable with the values estimated in the north of La Reunion island (i.e., 639 males/ha in September 2016 to 6,000 males/ha in December 2015) [87] and higher than estimates based on MRR experiments conducted in Rome in September 2009 (i.e., 135 mosquito/ha with regression and 294 m/ha with Fisher-Ford) [66]. Higher population size values (18,750–68,663 male/ha) are estimated by both statistical methods applied (despite GLM provided greater values than Fisher-Ford equation), under the assumption of a 50m dispersal. Although population size estimates are to be taken with caution as they are affected by the assumptions made (e.g., on dispersal) and by the methodological approaches applied (e.g., releases of males or females, recapture method, statistical

method), the very high number of *Ae. albopictus* individuals estimated on Procida confirms the capability of this species to reach very high population densities with detrimental effect on the quality of life of residents and tourists and, hence, the overall tourism-centred economy of the island.

### Dispersal, mean distance travelled and survival rates of sterile *Ae. albopictus* males

By applying the MRR technique, we observed a short-range dispersal of irradiated males, likely favoured by the uniform landscape of Procida with plenty of resting and swarming sites for adult males. However, the <60 m mean daily distance travelled by the males is in the range of values estimated for non-irradiated *Ae. albopictus* males in Switzerland [88] and La Reunion Island [87], and lower than values estimated for *Ae. albopictus* females in Italy [61,89] and Switzerland [88]. This suggests that the flying capacity of the released males is not affected by manipulations, irradiation and transportation and highlights the need of planning multiple short-distance ground releases and/or aerial uniform releases within a future SIT intervention program.

We observed a daily survival estimates of irradiated males ranging between 0.80 and 0.95, which are in the range of estimates for not-irradiated males in Missouri (USA) (survival rate = 0.77) [90]; Reunion Island (survival rate = 0.90 [87] and 0.94–0.95 [91]) and Switzerland = 0.88 [88]. These survival rates lessen the concern of excessive field mortality being a critical bottleneck for the success and sustainability of a SIT intervention [92]. In addition, this suggests that not only the dispersal capacity, but also the longevity of the released males is not affected by manipulations, irradiation and transportation, supporting the potential effectiveness of SIT interventions based on male production and sterilization in centralized facilities, even if distant from intervention area.

## Conclusions

The multifaceted approach described in this paper represents a guide for the identification of areas to test the cost-effectiveness and the challenges of mosquito control programs, integrating conventional (e.g., larval source removal and larvicide treatments) and innovative (e.g., SIT and/or IIT) control methods in Mediterranean Europe and in other temperate regions. Our approach could be used as a reference to design similar studies to evaluate conventional/innovative mosquito control methods and to improve community-engagement strategies for mosquito control.

In addition, results obtained represent an excellent endorsement of Procida Island as a site for future experimental trials. The ecological characteristics of the island, the high human and *Ae. albopictus* densities, the positive attitude of the resident population in being active parts in innovative mosquito control projects and the acquired knowledge on *Ae. albopictus* spatial and temporal distribution provide the ground for evidence-based planning of the interventions and for the assessment of their effectiveness. Finally, our results highlight the value of creating synergies between research groups, local administrators, and citizens for affordable monitoring (and, in the future, control) of mosquito populations.

## Supporting information

**S1 Fig. Procida island.** A) Procida Island map. Yellow box shows the extent of map showed in Fig 5, red box shows the extent of map showed in S5 Fig. The base layer of the geographic background map has been sourced from an open maps access (https://glovis.usgs.gov/app). B) Procida island satellite picture (https://eol.jsc.nasa.gov/SearchPhotos/). In both

representations of the island, a clear uniformity in the territory organization is visible. Most of the Procida territory is organized in private properties with residential buildings surrounded by green areas, including gardens with ornamental flowers, vegetable cultivations and/or orchards with citrus plants and family-type farming of chickens and rabbits. The only exception is represented by the Vivara natural reserve, which is inhabited by humans and connected with the main island by a concrete bridge (150m long).
(TIF)

**S2 Fig. Workflow of the community engagement approach utilized during the 4 years of activities on Procida island.** (*) MoU = Memorandum of Understanding. (**) ZanzaMapp is a mobile app for mosquito monitoring (https://www.zanzamapp.it/) [93] that was tested on Procida island during September 2016. (***) ERN = European Research Night. The public activities were organized on Procida island in the frame of the MEETmeTONIGHT project (http://www.meetmetonight.it/) funded by EU.
(TIF)

**S3 Fig. Volunteers involved during the three phases of the research program on Procida and Vivara Islands.** All the people (co-authors of the present manuscript and Procida volunteers) present in this figure gave their written consent to be photographed and to have their images published under a creative commons license.
(TIF)

**S4 Fig. Position of the ovitraps on Procida Island.** The position of the 101 ovitraps utilized for temporal (red) and spatial (red and green) analyses are reported. The base layer of the geographic background map has been sourced from an open maps access (https://glovis.usgs.gov/app).
(TIF)

**S5 Fig. Positions of ovitraps and BG-traps on Vivara Island.** Purple circles represent the position of the ovitraps. Black circles represent the position of BG-sentinel traps. Red circles indicate the position of two identified breeding sites, in both cases represented by ancient cisterns for the collection of rainwater. The base layer of the geographic background map has been sourced from an open maps access (https://glovis.usgs.gov/app).
(TIF)

**S6 Fig. GAMM model validation.** Upper right panel: Pearson's residuals versus fitted values. Upper left panel: Pearson's residuals versus temperature. Middle right panel: Histogram of % of zeroes obtained by simulating 10000 databases, the red dot represents the observed % of zeroes. Middle left panel: Pearson's residuals versus fitted date of collection. Lower right: Pearson's residuals versus ovitraps. Lower left: Variogram of Pearson's residuals. Autocorrelation function of each ovitrap time series did not show serious violation of independence.
(TIF)

**S7 Fig. Assessment of GAMM model fit.** Left panel: On the x axis the conditional predictive ordinate (CPO) which represents the posterior probability of observing that observation when the model is fit using all data except that one. On the y-axis the observation. Right panel: the frequency distribution of the probability of a new value to be lower than the actual observed value (Bayesian p-value).
(TIF)

**S8 Fig. Example of cloud variogram for week 29.** Each point on the plot represents a couple of two locations separated by a distance vector in 2D spatial domain (x axis) and having a

semivariance value reported on y axis.
(TIF)

**S9 Fig. Experimental semivariograms of total eggs/ovitrap/week on Procida.** Each point represents the average value of semivariance of couple of locations belonging to the same lag. This is called semivariogram and it is the main tool in geostatistics to discover the existence of spatial structure in the data. It is used to inform the interpolation by kriging.
(TIF)

**S10 Fig. Pattern of *Ae. albopictus* eggs/ovitrap/week and temperature on Procida Island in 2016 (upper panel) and 2017 (lower panel).** On the x-axis the date. The black solid lines represent the GAMM posterior predictive mean value of eggs in ovitrap (left y-axis), the shaded areas represent their 95% credible intervals. The red solid line represents temperatures (right y-axis). In the upper panel the black solid line (the posterior mean) is estimated from observed data (2016).
(TIF)

**S1 Text. Equations for mean distance travelled computation.**
(DOCX)

**S2 Text. Equations for expected recapture rate calculation.**
(DOCX)

**S1 Table. Data and coordinates for the temporal, spatial and MRR analyses.**
(XLS)

**S2 Table. Pearson correlation coefficients between ovitrap egg numbers in spatial analysis, terrain parameters derived from digital terrain model (dtm) and sea distance.**
(XLS)

## Acknowledgments

We are deeply grateful for the invaluable help of the Procida Major Raimondo Ambrosino and the municipal counsellors Rossella Lauro and Titta Lubrano. We are very grateful to the Procida volunteers Davide Zeccolella, Luigi "Corecane" D'Orio, Cesare Buoninconti, Amedeo Schiano, Michele Meglio, Alberto Salvemini, Marilena Scotto D'Apollonia, Max Noviello, Anna and Antonio Amalfitano, Emanuela Coppola, Biagio and Isa Coppola, Angela and Pasquale Lubrano, Giulia and Angelo Salvemini, Claudia Riccio and Antonietta Pagano, who shared their time to support this project. We thank Franco e Maria Costagliola and Pina Cuccurullo for the hosting of many students and researchers in their houses during the years of the project. We thank Nella Scotto for the assistance during the surveys on the island. We greatly thank Proff. Luciano Gaudio and Serena Aceto for their encouragement and support during these years of study on the island. We thank the students at the University of Naples Federico II involved in this project: Brunella Bozzi, Angela Meccariello, Rita Colonna, Antonia Fiore, Claudia Ascione, Daniela Carannante and Antonio Marino. We thank all the Procida citizens and tourist accommodations (Hotel Riviera, Hotel La Torre, Hotel Savoia, Camping Punta Serra, Camping Vivara) that granted us access to their private properties for mosquito monitoring activities. We thank prof. Franz Iandolo, prof. Veronica Nasti and the students of the Naples Academy of Fine Arts, NTA (New Technologies of Art) for their support during the communication campaign in 2018. We thank Pasquale Raicaldo for his support in reporting on national newspaper about our project progresses on the Island. We thank the Comitato di Gestione Isola di Vivara for the authorization to access the reserve. We are deeply grateful

to Rui Cardoso Pereira, Jeremy Bouyer, Kostas Bourtzis, Marc Vreysen and Jorge Hendrichs of the Joint FAO/IAEA Division of Nuclear Techniques in Food and Agriculture, Wien, Austria, for their help and support. We thank Jeremy Bouyer, Vincenza Colonna, Prof. Filiberto Cimino, Prof. Luciano Gaudio and Prof. Serena Aceto for their critical reading of the manuscript.

## Author Contributions

**Conceptualization:** Beniamino Caputo, Giuliano Langella, Romeo Bellini, Alessandra della Torre, Marco Salvemini.

**Data curation:** Giuliano Langella, Marco Salvemini.

**Formal analysis:** Marco Salvemini.

**Funding acquisition:** Marco Salvemini.

**Investigation:** Beniamino Caputo, Giuliano Langella, Valeria Petrella, Chiara Virgillito, Mattia Manica, Federico Filipponi, Marianna Varone, Pasquale Primo, Arianna Puggioli, Costantino D'Antonio, Luca Iesu, Liliana Tullo, Ciro Rizzo, Annalisa Longobardi, Germano Sollazzo, Maryanna Martina Perrotta, Miriana Fabozzi, Fabiana Palmieri, Roberto Rosà, Alessandra della Torre, Marco Salvemini.

**Methodology:** Beniamino Caputo, Valeria Petrella, Mattia Manica, Marianna Varone, Marco Salvemini.

**Project administration:** Marco Salvemini.

**Resources:** Giuseppe Saccone.

**Supervision:** Valeria Petrella, Marco Salvemini.

**Validation:** Marco Salvemini.

**Visualization:** Mattia Manica, Marco Salvemini.

**Writing – original draft:** Beniamino Caputo, Giuliano Langella, Alessandra della Torre, Marco Salvemini.

**Writing – review & editing:** Mattia Manica, Romeo Bellini, Giuseppe Saccone, Roberto Rosà, Marco Salvemini.

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
