## [Decision Letter · Decision Letter 0]

9 Apr 2021

Dear Prof. Salvemini,

Thank you very much for submitting your manuscript "Aedes albopictus bionomics in Procida Island, a promising Mediterranean site for the assessment of innovative and community-based integrated pest management methods." for consideration at PLOS Neglected Tropical Diseases. As with all papers reviewed by the journal, your manuscript was reviewed by members of the editorial board and by several independent reviewers. In light of the reviews (below this email), we would like to invite the resubmission of a significantly-revised version that takes into account the reviewers' comments. 

We cannot make any decision about publication until we have seen the revised manuscript and your response to the reviewers' comments. Your revised manuscript is also likely to be sent to reviewers for further evaluation.

Sincerely,

Adly M.M. Abd-Alla, Prof asso.

Associate Editor

Amy Morrison

Deputy Editor

Reviewer's Responses to Questions

**Key Review Criteria Required for Acceptance?**

**Methods**

-Are the objectives of the study clearly articulated with a clear testable hypothesis stated?

-Is the study design appropriate to address the stated objectives?

-Is the population clearly described and appropriate for the hypothesis being tested?

-Is the sample size sufficient to ensure adequate power to address the hypothesis being tested?

-Were correct statistical analysis used to support conclusions?

-Are there concerns about ethical or regulatory requirements being met?

Reviewer #1: Minor revision - additional experiments are not required, but methods section needs improving. An additional experiment assessing the impact of SIT on population abundance, biting pressure, etc would significantly strengthen this paper.

Reviewer #2: The objectives of this study were to use a community-based approach to collect baseline data about mosquito bionomics on the island; while no testable hypothesis was stated, it is apparent that the successful collection of biomic data would validate the feasibility of this approach.

A primary source of data for this manuscript is egg collection from ovitraps, and as such it is crucial that the collection method allowed for accurate egg quantification. In the methods, it is stated that the cylindrical traps (12cm diameter) have an overflow hole at 8cm, yielding a capacity of just under 1L before overflow, and 600mL of water are added before weekly checks-- this starting volume is calculated to come up to ~5.5cm from bottom of the trap, or ~2.5 cm from overflow hole. However, no confirmation is provided that water overflow did not result in the loss of eggs; with 80cm of rainfall per year (methods: study site), several cm of rainfall and subsequent egg loss in a given week would appear to be quite feasible. The methods used to prevent egg loss, accounting for any overflow, and/or confirmation that such loss did not occur should be thoroughly discussed.

Several critiques of the MRR methods are raised:

1) The quantity of fluorescent powder applied to the mosquitoes is not indicated; Culbert et al (PMC7158013) show that Ae albopictus longevity is affected by this marking process at all but the lowest doses tested, which should be accounted for in the analysis.

2) I was unable to locate any information on the fluorescent powder used based on the information provided in the manuscript: while fluorescent dust has been reported to be detectable on mosquitoes for upwards of one month (e.g. PMC71558013), a valuable addition to the experiment would be to have a cage of marked mosquitoes remain outside at the release site (or equivalent environment in regards to wind, rain, humidity, etc) to ensure that dusting was adequate to ensure detection at 13 days. Alternatively, the authors should indicate why they are confident that the dusting was sufficient for detection throughout the study.

3) The use of human landing catches is unexpected given the collection of males, which are not attracted to humans. Please explain the rationale in the methods.

Reviewer #3: In general, my main methodological objection is the apparent lack of geographical and bionomical descriptions: breeding sites are not studied even though they would compete with ovitraps, and land cover is used as a covariate not influencing the mosquito distribution, which could be quite surprising in a less uniform landscape. These topics are somewhat adresses in discussion, but i think they should be introduced from the beginning, very especially about the particular urban configuration of the majority of the land. Taking a view of the island on Google maps is clarifying, I think this kind of graphical information would be necessary from the beginning. 

L191 - Is really 2.83 inh/Ha a "very high" (human) population density? I feel this is not to be considered really high in comparison to other urbanized areas.

L207 and others - I feel (as the authors actually highlighted in discussion) that the ovitrapping data are a bit poor, both in seasonality (first populations appearing late April whereas the ovitrapping only started mid- April) but also suffering from a single-year sampling. 

L226 - Is it possible to explain which were the other public activities?

L241 - What is the male separation efficiency of this method? Was the number of released females assessed?

L273 - It is quite surprising the use of HLC captures for males since this is a method traditionally used to capture biting females. This should be discussed - if a general, manual male collection is involved, another acronym should be used (and screened areas, such as vegetation, described).

Reviewer #4: The methods are well designed addressing the objectives of the study, and the statistical analyses is appropriate. Language editing, spelling mistakes need to be corrected.

**Results**

-Does the analysis presented match the analysis plan?

-Are the results clearly and completely presented?

-Are the figures (Tables, Images) of sufficient quality for clarity?

Reviewer #1: Major revision - there was a great deal of method repetition in the results section and results were not clearly communicated. The section needs to be restructured with more consideration given to the key results from each piece of the study.

Reviewer #2: While the analysis presented match the overall goals of this project, several issues with the data presentation arise. Minor modifications are suggested in the “Editorial and Data Presentation Modifications” section, while more significant requests are listed here:

Figure 1: 

Since the results in figure 1 are predictions based on a model based on collected data, it would be useful to see the collected data on which the model was based as one panel, and the model predictions (currently the left panel) as a second panel

Figure 2:

The text refers to Fig 2 only briefly and when describing the relationship between predicted and observed values, and should more fully describe the 26 panels and the relevance of the model being accurate across all sample locations. Further, it is not clear whether the Pearson’s correlation indicating a positive correlation are for any of the 26 sample locations, all of them taken collectively, or the data in figure S3.

Figure 3:

It is recommended that the authors consider revising this figure to more clearly convey the relationship between temperature and egg deposition, by combining the independent (temperature) and dependent (eggs) variables onto a single graph. Two panels for the two years are suggested since the years are fundamentally different, in that 2016 is based on observed data, and 2017 is a prediction. 

Figure 4:

The shades of blue and circle size are unclear: the description of “week” above is difficult to understand, as each panel appears to refer to a single week. The difference between color and size of circle is not indicated; if they are redundant, this redundancy should be eliminated or described. The numbers (e.g. 100, 500) have no units, I believe this refers to eggs/ovitrap but this should be indicated. Finally, all four panels should use the same scale, since they are quite similar, and the top right panel should have the color/size legend flipped to match the rest. In all, this figure needs substantial work to be comprehendible by the reader.

Reviewer #3: Most results are conforming to what one could expect. Spatial analysis would benefit from an analysis on the distribution of natural breeding sites - similarly, in Vivara reserve breeding sites should be identified (are Tiger mosquitoes present there in high densities coming from mainland?).

Table 4 - when expressing estimates of "natural" Aedes albopictus populations I guess these are only male populations, like the MRR ones.

Reviewer #4: The analysis of the study is sufficient and meet the analysis plan, the results are also clearly presented, spelling and punctuations need to be revised.

**Conclusions**

-Are the conclusions supported by the data presented?

-Are the limitations of analysis clearly described?

-Do the authors discuss how these data can be helpful to advance our understanding of the topic under study?

-Is public health relevance addressed?

Reviewer #1: Major revision - The discussion section only weakly discusses the paper’s findings in context of other literature. It is clear that there is value to the work that was conducted, but it is not framed and discussed appropriately throughout the discussion section. I suggest bulking that section up and really connecting your research to other’s. It will have the most impact when it is considered amongst the broader database of research on the topic.

Reviewer #2: The conclusions are supported by the data presented, and the authors make excellent points in the Discussion regarding the applicability of this work to future pest management efforts.

Reviewer #3: Discussion and conclusions are adequate as they also underline (though for the first time across all the manuscript) some of the methodological weaknesses: egg monitoring for only one season, assumptions on the MRR approach. I would add here a mention on the breeding sites distribution, maybe across L687-L690 where a massive and uniform presence of Ae.albopictus is claimed. Most of this, however, should not be placed on discussion but in Material and Methods because it is a description of the study area which should come forefront.

Reviewer #4: The conclusion Final conclusions of the paper are too general, some sentences suits the discussion rather than the conclusion. It needs to be more precise and concise. recommendations for similar future studies favored to be added.

**Editorial and Data Presentation Modifications?**

Reviewer #1: Minor revision - primarily editorial. Most figures are displayed in a clear way that is easy to understand.

Reviewer #2: I don’t have a strong modeling background and thus cannot comment on the appropriateness of the models or presentation; these may be of limited interest to the average reader, and the specifics of the models may be better suited for supplemental materials.

In the methods (citizen science section), the period of Sept 2015-Sept 2018 is indicated as a four-year span (that period is only three years).

The discussion refers to the second survey taking place in 2018, while the results section lists 2019; please align to the correct year. 

Figure S8 does not contribute appreciably to the manuscript and should be removed.

Figure 1:

-Requires units in the x-axis of both panels

Figure 4:

-Requires units for the y-axis of the upper panel.

Table 3:

Requires a unit for all numbers: meters?

Reviewer #3: L99 - Spanish event was not Chikungunya but Dengue, need to change that

L106 - Using mainly chemical pesticides should perhaps be framed as an Italian topic. In several other EU countries the Tiger mosquito is being dealt with by massive microbian larviciding, which additionally should merit some discussion in this paragraph.

L112 - On the description of SIT or IIT I would prefer to always use the adjective "experimental" complementary control methods. These techniques exist and are being tested since decades on mosquitoes but haven't yet been on practical use.

Reviewer #4: The paper is presenting an implementation research towards Aedes albopictus the principal vector of arboviruses and dengue control. The introduction needs a lot of language editing. After some minor revision I think the paper will be suitable to publish in your precious journal.

**Summary and General Comments**

Reviewer #1: To the authors,

Thank you for contributing your work to PLOS NTD. It is clear that an immense amount of work went into this project and it was truly a collaborative effort. I hope that my comments below and in my marked pdf aid you in improving the manuscript. 

There is certainly scientific rigor in the research you conducted, but it’s communication in this manuscript needs significant improvements. If this is done, I have no doubt that it will improve the clarity and readability of the paper and allow it to reach a broader scientific community. 

In general, I believe the tone of the paper is at times too casual. The authors did a great job of providing detail and resources, but phrases like “Thanks to their help” and other similar phrases should not be used. There are more formal ways of making these points that are appropriate for scientific papers. 

I also think the authors would benefit from using more concise language wherever possible. There are many extra words used throughout the paper that can be eliminated. The word very was used a lot, for example. I would suggest having a fresh set of eyes read through before submitting the revision. It just needs tightening up. 

At some points, I feel the level of detailed provided exceeds what is traditionally included in a manuscript. Much of that information can be removed and simplified. 

The results section feels very unclear. I don’t feel like the major finding from analysis or trends in the data were discussed in an understandable way. It’s difficult to read as currently written and there are pieces that are more fitted for other sections of the paper. I have tried to point some examples out, but it can be restructured to benefit the readability and clarity of the paper.

The discussion section only weakly discusses the paper’s findings in context of other literature. It is clear that there is value to the work that was conducted, but it is not framed and discussed appropriately throughout the discussion section. I suggest bulking that section up and really connecting your research to other’s. It will have the most impact when it is considered amongst the broader database of research on the topic. 

Overall, I would like to repeat my suggestion to have a fresh set of eyes read through the paper. It needs some significant restructuring and can be formalized and trimmed down significantly. Despite these needs for the paper, I still believe the work to be valuable and worthy of publication, but only with these significant improvements.

My specific comments are included in the pdf but are by no means exhaustive or all inclusive. Consider my comments in context to the entire paper, not just the pieces I called attention to. Best of luck with revisions.

Reviewer #2: This manuscript provides a useful description of population characteristics of the Ae. albopictus mosquito on Procida, Italy, and the utility of this small Italian island for pest management studies. Further, the authors went to great lengths to involve the local community in the execution of this research, and have presented a convincing model for a community-based approach to mosquito management. The authors show that the Ae. albopictus egg dynamics are temperature-dependent, weighted towards the end of the summer, are uniformly distributed across the island, and that male dispersal is generally <60m and daily survival is around 0.8-0.95.

While the findings are novel for this particular location, and this island provides a useful tool for future pest management studies, none of the findings add substantial knowledge to the field of vector biology, as all results have been previously reported for Ae. albopictus in other locations, including mainland Italy (e.g. PMC7954105 for temperature-dependence of egg laying in Italy and weight towards the end-of-summer, PMC5669009 for distribution of mosquitoes on Italian islands, PMC6371565 for similar daily survival and dispersal distances).

Some concerns regarding the methods data presentation would warrant revision prior to suitability of this study for publication.

Finally, the language and grammar of the manuscript requires extensive revision to be publication-worthy, and I advise that the authors work with a writing coach to ensure that the manuscript complies with the PLOS NITD requirement to be presented in standard English. Frequent errors in word choice, sentence structure, and mistakes in singular/pluralization as well as un-defined abbreviations require attention. Substantial clarification requests for most figures, such as axis units and description of gradients, emphasize the need for additional editorial work.

Reviewer #3: This is a study dealing with the description of an Asian Tiger mosquito population on an island location which is really adequate as a test field for SIT and any other control technique. The work is well designed in view of the aimed goals, though the experimental base data is in my opinion somewhat too light to support conclusions drawn from quite advanced modelling statistics. In general, no big surprises over what was to be expected, resulting anyway in interesting population data and conclusions. 

Citizenship involvment has been sought and obtained from a number of individuals to help in trapping and is highllighted by the authors as a crucial asset in future operations but is not thoroughly described - some notes on what was carried out in this area would be welcome, especially regarding other awareness actions, furthermore considering the survey results were quite poor in terms of user engagement. 

We believe this work can be published with some extra clarification.

Reviewer #4: The paper revealed a nice piece of implemented research. The author summary and introduction included many misspellings, imperfect and unclear sentences, from my perspective extensive language editing need to be conducted prior publication.

PLOS authors have the option to publish the peer review history of their article (what does this mean?). If published, this will include your full peer review and any attached files.

Reviewer #1: No

Reviewer #2: No

Reviewer #3: No

Reviewer #4: No
---

## [Decision Letter · Decision Letter 1]

18 Jul 2021

Dear Prof. Salvemini,

Thank you very much for submitting your manuscript "Aedes albopictus bionomics data collection by community participation in Procida Island, a promising Mediterranean site for the assessment of innovative and community-based integrated pest management methods." for consideration at PLOS Neglected Tropical Diseases. As with all papers reviewed by the journal, your manuscript was reviewed by members of the editorial board and by several independent reviewers. The reviewers appreciated the attention to an important topic. Based on the reviews, we are likely to accept this manuscript for publication, providing that you modify the manuscript according to the review recommendations. 

Sincerely,

Adly M.M. Abd-Alla, Prof asso.

Associate Editor

Amy Morrison

Deputy Editor

Reviewer's Responses to Questions

**Key Review Criteria Required for Acceptance?**

**Methods**

-Are the objectives of the study clearly articulated with a clear testable hypothesis stated?

-Is the study design appropriate to address the stated objectives?

-Is the population clearly described and appropriate for the hypothesis being tested?

-Is the sample size sufficient to ensure adequate power to address the hypothesis being tested?

-Were correct statistical analysis used to support conclusions?

-Are there concerns about ethical or regulatory requirements being met?

Reviewer #1: Yes, the authors have made significant improvements to the text. The methods are more appropriately described now.

Reviewer #2: The critiques I raised about the methods were adequately addressed by the authors

Reviewer #4: The methods were improved and corrected following reviewers comments. The methods are well designed addressing the objectives of the study, and the statistical analyses is appropriate. Language editing, spelling mistakes need to be corrected.

**Results**

-Does the analysis presented match the analysis plan?

-Are the results clearly and completely presented?

-Are the figures (Tables, Images) of sufficient quality for clarity?

Reviewer #1: Yes, this section has also been improved and the information is completely presented.

Reviewer #2: The requested edits and clarifications to the results section have been appropriately incorporated into the revised manuscript.

Reviewer #4: The results are clearly presented. Table 1 the title is not clear, it needs to be more clear (ex: is it a community survey?), it is also better if you put the title up in the table and the legend in bottom.

The tables are not in the same design and layout, please correct.

**Conclusions**

-Are the conclusions supported by the data presented?

-Are the limitations of analysis clearly described?

-Do the authors discuss how these data can be helpful to advance our understanding of the topic under study?

-Is public health relevance addressed?

Reviewer #1: Conclusions are concise and represent the major findings of the paper.

Reviewer #2: No further recommendations

Reviewer #4: The conclusion are supported by the presented results, however the limitations and recommendations to the future research in the filed were not sufficiently addressed.

**Editorial and Data Presentation Modifications?**

Reviewer #1: Could still use some minor grammar editing, but is otherwise much improved from the original submission.

Reviewer #2: No further recommendations

Reviewer #4: I think the authors need to modify the manuscript title to be more strong, clear and represent the study accurately.

**Summary and General Comments**

Reviewer #1: The authors made significant improvements to the paper that improve its clarity and readability. Identifying all changes without the track changes document was difficult, but the paper is more logically presented now.

Reviewer #2: The authors have substantially revised the manuscript to be readable, and have addressed the concerns that I listed in my original review. The article is now recommended for publication.

Throughout the manuscript, including the title, "in Procida island" is used when "on Procida island" may be more appropriate.

The authors should not exclusively use the pronouns 'he' and 'his' when responding to reviewer comments, but rather use statements like "we thank the reviewer for their comment".

Reviewer #4: After the authors consider all the comments the manuscript will be ready to publish in the journal.

I think the authors need to modify the manuscript title to be more strong, clear and represent the study accurately.

PLOS authors have the option to publish the peer review history of their article (what does this mean?). If published, this will include your full peer review and any attached files.

Reviewer #1: No

Reviewer #2: No

Reviewer #4: No

Figure Files:

Data Requirements:

Reproducibility:

References

---

## [Editor Report · Decision Letter 2]

2 Aug 2021

Dear Prof. Salvemini,

We are pleased to inform you that your manuscript 'Aedes albopictus bionomics data collection by citizen participation on Procida Island, a promising Mediterranean site for the assessment of innovative and community-based integrated pest managment methods.' has been provisionally accepted for publication in PLOS Neglected Tropical Diseases.

Best regards,

Adly M.M. Abd-Alla, Prof asso.

Associate Editor

Amy Morrison

Deputy Editor

---

## [Editor Report · Acceptance letter]

31 Aug 2021

Dear Prof. Salvemini,

We are delighted to inform you that your manuscript, "Aedes albopictus bionomics data collection by citizen participation on Procida Island, a promising Mediterranean site for the assessment of innovative and community-based integrated pest management methods.," has been formally accepted for publication in PLOS Neglected Tropical Diseases.

Best regards,

Shaden Kamhawi

co-Editor-in-Chief

Paul Brindley

co-Editor-in-Chief
